# C-Voting: Confidence-Based Test-Time Voting without Explicit Energy Functions

**Kenji Kubo,**[*]**Shunsuke Kamiya, Masanori Koyama, Kohei Hayashi, Yusuke Iwasawa & Yutaka Matsuo**
Graduate School of Engineering,
The University of Tokyo

## Abstract

Neural network models with latent recurrent processing, where identical layers are recursively applied to the latent state, have gained attention as promising models for performing reasoning tasks. A strength of such models is that they enable test-time scaling, where the models can enhance their performance in the test phase without additional training. Models such as the Hierarchical Reasoning Model (HRM) and Artificial Kuramoto Oscillatory Neurons (AKOrN) can facilitate deeper reasoning by increasing the number of recurrent steps, thereby enabling the completion of challenging tasks, including Sudoku, Maze solving, and AGI benchmarks. In this work, we introduce confidence-based voting (C-voting), a test-time scaling strategy designed for recurrent models with multiple latent candidate trajectories. Initializing the latent state with multiple candidates using random variables, C-voting selects the one maximizing the average of top-1 probabilities of the predictions, reflecting the model's confidence. Additionally, it yields $4.9\%$ higher accuracy on Sudoku-hard than the energy-based voting strategy, which is specific to models with explicit energy functions. An essential advantage of C-voting is its applicability: it can be applied to recurrent models without requiring an explicit energy function. Finally, we introduce a simple attention-based recurrent model with randomized initial values named ItrSA++, and demonstrate that when combined with C-voting, it outperforms HRM on Sudoku-extreme ($95.2\%$ vs. $55.0\%$) and Maze ($78.6\%$ vs. $74.5\%$) tasks.

## 1 Introduction

In recent years, reasoning has been recognized as crucial for achieving Artificial General Intelligence (AGI). Recurrent models, which repeat identical layers, have emerged as a promising approach to achieve the goal. A key advantage of recurrent models is that their performance can be enhanced at test time without additional training, a technique known as test-time scaling. The test-time scaling is typically realized in two ways: (1) increasing the number of inference recurrence steps, and (2) selecting a "good" trajectory from multiple candidates with random sampling, or *voting*. The method of increasing inference steps (1) has been investigated in (Anil et al., 2022) and other studies. The method of selecting "good" trajectories (2) has recently attracted attention in large language models (LLMs), particularly in the context of self-consistency (Wang et al., 2023) as well as search and decoding strategies (Yao et al., 2023; Zhou et al., 2022).

Recent recurrent models, such as the Hierarchical Reasoning Model (HRM) (Wang et al., 2025) and the Artificial Kuramoto Oscillatory Neurons (AKOrN) (Miyato et al., 2025), have become capable of solving complex tasks, including Sudoku and Maze by utilizing test-time scaling. Sudoku and Maze are regarded as benchmarks for reasoning because they require consistent logical inference under complex constraints. These are challenging tasks for conventional models, including leading LLMs as reported in (Wang et al., 2025). HRM mainly relies on increasing recurrence depth, whereas AKOrN introduces energy-based voting (E-voting) in addition to increasing the depth. E-voting samples multiple initial latent states and selects the final trajectory with the lowest energy, yielding large performance gains in Sudoku. It enables further improvement even in cases where performance improvement from increasing the number of recurrent steps is saturated. Specifically,

---

[*]kenji.kubo@weblab.t.u-tokyo.ac.jp

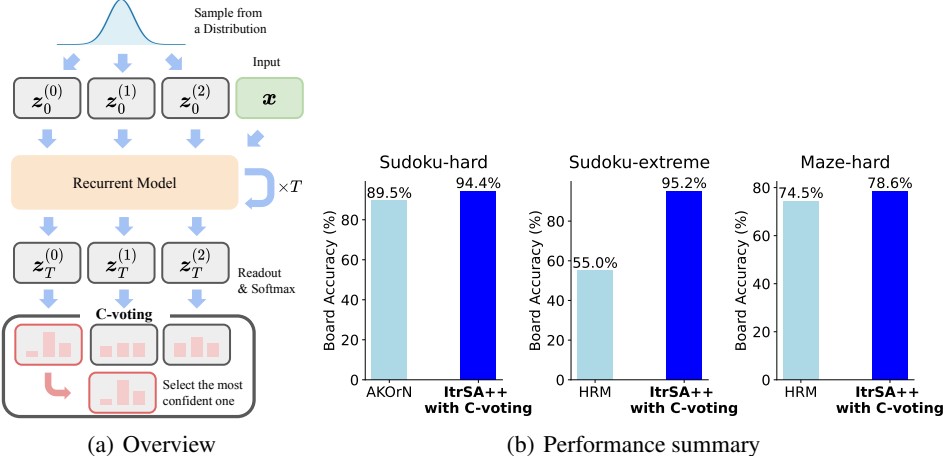

(a) Overview    (b) Performance summary

Figure 1: (a): The overview of confidence-based voting (C-voting). The candidates of the latent state are initialized by random variables, and the recurrent model updates the latent states from $t = 0$ to $t = T$ using input $x$. Calculating probabilities from the final states, C-voting selects the most confident prediction. (b): The performance comparisons of ItrSA++ with C-voting to other state-of-the-art models. The proposed method outperforms in Sudoku-hard, Sudoku-extreme, and Maze-hard tasks.

it has been reported that using $4096$ candidates resulted in an approximately $40\%$ increase in the board accuracy (Miyato et al., 2025). Although highly effective, it has the limitation that it cannot be used unless the energy function is explicitly defined. In most promising reasoning models, such as HRM and recurrent transformers (Geiping et al., 2025), the energy is not explicitly defined, making it impossible to use E-voting. This limitation raises natural questions: (RQ1) Can we facilitate a model-agnostic test-time voting strategy without explicit energy functions, and (RQ2) Does there exist a simple, lightweight recurrent architecture matching or surpassing state-of-the-art performance with C-voting?

In this work, we propose confidence-based voting (C-voting), a novel voting strategy at test time, which enhances performance on a broader range of recurrent models. C-voting only requires the recurrent models to start from randomly sampled initial values, and does not require explicit energy functions. The model initializes the latent states with random variables, updates them applying an identical model repeatedly, and performs inferences. With these sampled trajectories, the rule of C-voting is very simple: take the most confident one (Figure 1(a)). We measure the confidence of a trajectory through the average top-1 probability that the trajectory finally provides. We experimentally find that C-voting offers better performance than E-voting in the case of Sudoku-solving with AKOrN.

C-voting can be applied not only to existing models, but also to newly designed, more powerful recurrent models. As an example, we introduce ItrSA++, a simple recurrent model with randomized initial values. It outperforms HRM on Sudoku-extreme ($95.2\%$ vs. $55.0\%$) and Maze-hard tasks ($78.6\%$ vs. $74.5\%$), and AKOrN on Sudoku-hard task ($94.4\%$ vs. $89.5\%$) as shown in Figure 1(b). The model only requires about 3 million parameters, which is about one-ninth the number of parameters in HRM, and comparable to that of AKOrN. We also demonstrate that C-voting can be applied to HRM with C-voting. This represents that C-voting is a simple and portable test-time voting strategy that enables recurrent models to achieve state-of-the-art performance with significantly fewer parameters.

## 2 PRELIMINARIES

### 2.1 RECURRENT MODELS WITH RANDOMIZED INITIALIZATION

Recent works show that recurrent models, which apply identical layers iteratively, are a powerful and promising framework for a variety of complex reasoning tasks (Dehghani et al., 2019; Anil et al.,

2022; Geiping et al., 2025; Jaegle et al., 2021; Gladstone et al., 2025; Wang et al., 2025; Miyato et al., 2025; Saunshi et al., 2025; Darlow et al., 2025). Concretely, these recurrent models typically have the form of

$$\boldsymbol{z}_{i,t+1} = f(\boldsymbol{z}_{i,t}, \boldsymbol{x}_i; \boldsymbol{\theta}), \tag{1}$$

where $f$ is a neural network, $\boldsymbol{z}_{i,t} \in \mathbb{R}^{L \times C}$ the latent state at layer $t(\in \mathbb{N})$, $\boldsymbol{x}_i \in \mathbb{R}^{L \times C}$ the $i$-th input (e.g., image, Sudoku board with blanks) embedded in the latent space, and $\boldsymbol{\theta}$ the parameters. $L \in \mathbb{N}$ is the number of tokens, and $C \in \mathbb{N}$ is the embedding dimensions. For models that solve tasks with complex reasoning, such as Sudoku or Maze tasks, $f$ is oftentimes set to have a Transformer-based architecture (Vaswani et al., 2017; Dehghani et al., 2019; Saunshi et al., 2025). Note that the network $f(\cdot, \cdot; \boldsymbol{\theta})$ is identical for every $t$, hence equation 1 can be thought of as a time-invariant nonlinear dynamical system.

The behavior of the latent dynamics equation 1 depends on the initialization of $\boldsymbol{z}_{i,0}$, which possibly improves or deteriorates the model's performance. One strategy is to sample $\boldsymbol{z}_{i,0}$ from a probability distribution, e.g., a standard Gaussian distribution. Such randomized initialization can lead the model to learn path independence, or the convergence to the same final steady state (Anil et al., 2022), which helps the model to generalize better than initializing the dynamics with a fixed value, e.g., $\boldsymbol{z}_{i,0} \equiv 0$.

After repeating equation 1 from $t = 0$ to $t = T - 1$ $(T \in \mathbb{N})$, the final latent state $\boldsymbol{z}_{i,T}$ is passed to the readout module. In classification problems such as Sudoku, the readout module produces logits, which are subsequently converted to class probabilities by the softmax function or its variant.

Recurrent models with randomized initialization are beneficial not only in parameter efficiency but also in that they allow *test-time scaling*. Test-time scaling is a phenomenon where the model enhances its performance without further training, and is observed in a number of recurrent models (Gladstone et al., 2025; Geiping et al., 2025; Miyato et al., 2025; Hu et al., 2025). Test-time scaling can be realized in the following two ways: one is to run more iterations at inference than in training; the second is to select from the trajectories, which are initialized randomly, the "best" one based on a pre-determined criterion.

Although the term "test-time scaling" typically refers only to the concept of extending iterative steps, which has been studied extensively (Anil et al., 2022; Geiping et al., 2025), here we place our focus on the other concept, that is, selecting one trajectory from the sampled ones. We consider that this "sample & choose", or *voting* strategy, is as significant as extending iterative steps, as the voting strategy can compensate for the next two shortcomings of iterative step extension. First, the performance of a recurrent model cannot grow monotonously with extending iterative steps: the performance can saturate. Second, extending the iterative steps in the test phase requires long iterative computations and cannot be parallelized. The voting strategy works well with parallel computing and can enhance the performance even when it is saturated. Therefore, voting can enhance the reasoning ability to the point where the step extension cannot be solely reached.

## 3 RELATED WORK

### 3.1 RECURRENT MODELS WITH RANDOMIZED INITIALIZATION

As representative examples of the recurrent models with randomized initial values, we focus on HRM (Wang et al., 2025) and AKOrN (Miyato et al., 2025), as they demonstrate outstanding performance in logical tasks such as Sudoku and Maze.

**HRM** HRM (Wang et al., 2025) is a recurrent model that is gaining attention for its strong reasoning abilities for the relatively small size ($\approx$ 27 million parameters), reaching SoTA in Sudoku-extreme, Maze-hard, and ARC-AGI challenges (Chollet, 2019; Chollet et al., 2025). The latent dynamics is a two-fold model that has *higher* and *lower* order processing given as follows:

$$\begin{aligned} \boldsymbol{z}_{i,t+1}^{\mathrm{L}} &= f^{\mathrm{L}}\left(\boldsymbol{z}_{i,t}^{\mathrm{L}}, \boldsymbol{z}_{i,t}^{\mathrm{H}}, \boldsymbol{x}; \boldsymbol{\theta}^{\mathrm{L}}\right), \\ \boldsymbol{z}_{i,t+1}^{\mathrm{H}} &= \begin{cases} f^{\mathrm{H}}\left(\boldsymbol{z}_{i,t}^{\mathrm{H}}, \boldsymbol{z}_{i,t}^{\mathrm{L}}; \boldsymbol{\theta}^{\mathrm{H}}\right) & \text{if } t \equiv 0 \pmod{\tau} \\ \boldsymbol{z}_{i,t}^{\mathrm{H}} & \text{otherwise} \end{cases}, \end{aligned} \tag{2}$$

where $\tau$ is an update cycle for higher-level layers. The superscripts H and L respectively represent higher and lower order processings. This two-fold structure was first proposed as the main breakthrough but later shown to have a limited effect on the performance (Team, 2025). HRM makes use of other techniques to boost its performance, such as one-step gradient approximation (taking its origin from Deep Equilibrium Model (Bai et al., 2019)) or adaptive computational time (ACT).

**AKOrN** AKOrN (Miyato et al., 2025) is another recurrent model that shows SoTA performance in the Sudoku-hard task (Palm et al., 2018). AKOrN utilizes the $C$-dimensional Kuramoto oscillators (Kuramoto, 1975; Lipton et al., 2019) as the latent dynamics and given as:

$$\Delta \boldsymbol{z}_{i,t} = \Omega(\boldsymbol{z}_{i,t}) + \mathrm{Proj}_{\boldsymbol{z}_{i,t}} \left( \boldsymbol{x}_i + \boldsymbol{J}\left(\boldsymbol{z}_{i,t}\right) \right), \tag{3}$$

$$\boldsymbol{z}_{i,t+1} = \Pi \left( \boldsymbol{z}_{i,t} + \eta \Delta \boldsymbol{z}_{i,t} \right). \tag{4}$$

Here, all rows of $z_{i,t}$ are confined on the unit hypersphere (thus have norm one). $\Omega : \mathbb{R}^{L \times C} \to \mathbb{R}^{L \times C}$ is a linear map that rotates each token on the hypersphere, and thus is represented as a set of skew-symmetric matrices. $\boldsymbol{J} : \mathbb{R}^{L \times C} \to \mathbb{R}^{L \times C}$ is the connectivity strength between each token pair, and is implemented using self-attention. The projection onto the tangent space $\mathrm{Proj}$ and normalization onto the hypersphere $\Pi$ are applied token-wise.

## 3.2 TEST-TIME SCALING WITH E-VOTING

E-voting is a test-time voting strategy introduced in (Miyato et al., 2025). E-voting can be applied to the recurrent model equipped with an energy (scalar) function —i.e., the recurrent models where a scalar function $E : \mathbb{R}^{L \times C} \to \mathbb{R}$ exists, such that

$$\boldsymbol{z}_{i,t+1} = f(\boldsymbol{z}_{i,t}, \boldsymbol{x}_i; \boldsymbol{\theta}) = \boldsymbol{z}_{i,t} - \alpha \nabla_{\boldsymbol{z}} E(\boldsymbol{z}_{i,t}; \boldsymbol{\theta}), \quad \alpha \in \mathbb{R}_+. \tag{5}$$

This dynamics proceeds in the direction in which the energy (expectedly) decreases when $\alpha$ is small enough. For example, AKOrN has a closed-form energy function[1], whereas energy-based transformer (Gladstone et al., 2025) explicitly models an energy using a Transformer-based architecture. As the model learns the dynamics to decrease the energy function, a smaller energy value is expected to reflect higher accuracy. E-voting is based on this assumption, and proceeds as follows. We first sample initial states $\boldsymbol{z}_{i,0}$ for $K \in \mathbb{N}$ times from a probability distribution, which we denote by $\{\boldsymbol{z}_{i,0}^{(k)}\}_{k \in [1,K]}$. We then select the sample with the lowest energy at the final step $t = T$, i.e.,

$$k^* = \arg\min_k E(\boldsymbol{z}_{i,T}^{(k)}; \boldsymbol{\theta}). \tag{6}$$

This voting strategy is effective in the Sudoku-solving task; in fact, AKOrN achieves about a 40% improvement with E-voting.

Although E-voting is an effective voting strategy, it is not available in many models since the energy function is not explicitly obtained. For example, in recurrent models with a residual connection

$$\boldsymbol{z}_{i,t+1} = \boldsymbol{z}_{i,t} + g(\boldsymbol{z}_{i,t}; \boldsymbol{\theta}), \tag{7}$$

$g$ has to be written as the gradient of some scalar function, which does not exist in general.

## 4 CONFIDENCE-BASED VOTING

We describe the detailed procedure of the proposed voting method, C-voting. C-voting, as the name indicates, is a simple voting strategy applicable at test time. Concretely, C-voting applies the following steps to the trained model. We first sample initial states $\boldsymbol{z}_{i,0}$ for $K \in \mathbb{N}$ times from a probability distribution, which we denote by $\{\boldsymbol{z}_{i,0}^{(k)}\}_{k \in [1,K]}$. From these sampled initial states, we update the latent state with a neural network $f$ by equation 1. Reading out the final state $\boldsymbol{z}_{i,T}^{(k)}$ as logits, we obtain the probability of the class $j$ at the position $l$ of $k$-th candidate as

$$P_{j,l}(\boldsymbol{z}_{i,T}^{(k)}) = \mathrm{Softmax}\left(\mathrm{Readout}(\boldsymbol{z}_{i,T}^{(k)})\right)_{j,l}, \tag{8}$$

---

[1] Note that equation 5 does not strictly hold when $\boldsymbol{J}$ in equation 3 is asymmetric. Nevertheless, E-voting is shown to be effective for AKOrN in such a case.

where the softmax function is applied in the dimension of classes. The position $l$ indicates the location at which the probability is computed, e.g., a cell in Sudoku. Then, the top-1 probability at the position $l$ is

$$\hat{P}_l(\boldsymbol{z}_{i,T}^{(k)}) = \max_j P_{j,l}(\boldsymbol{z}_{i,T}^{(k)}). \tag{9}$$

The average of top-1 probabilities over all positions reflects the confidence of the model. Therefore, we define the model's confidence for the $i$-th input and the $k$-th candidate as

$$C_i^{(k)} = \frac{1}{|\mathcal{L}|} \sum_{l \in \mathcal{L}} \hat{P}_l(\boldsymbol{z}_{i,T}^{(k)}), \tag{10}$$

where $\mathcal{L}$ denotes the set of positions to be predicted, e.g., indices of blank cells in Sudoku, and $|\cdot|$ denotes the number of elements in the set. We select the index of the most confident candidate by

$$k_i^* = \arg\max_k C_i^{(k)}. \tag{11}$$

Thus, the prediction with the C-voting for input $i$ at position $l$ is obtained by

$$\hat{y}_{i,l}^* = \arg\max_j P_{j,l}(\boldsymbol{z}_{i,T}^{(k_i^*)}). \tag{12}$$

If the model is well calibrated, the following approximation holds

$$\Pr[y_{i,l} = \hat{y}_{i,l}^{(k)} | \hat{P}_l(\boldsymbol{z}_{i,T}^{(k)})] \simeq \hat{P}_l(\boldsymbol{z}_{i,T}^{(k)}), \tag{13}$$

where $\hat{y}_{i,l}^{(k)} = \arg\max_j P_{j,l}(\boldsymbol{z}_{i,T}^{(k)})$. Using equation 13, we obtain the index of the candidate that maximizes average accuracy across all positions by

$$\arg\max_k \frac{1}{|\mathcal{L}|} \sum_{l \in \mathcal{L}} \Pr[y_{i,l} = \hat{y}_{i,l}^{(k)} | \hat{P}_l(\boldsymbol{z}_{i,T}^{(k)})] \simeq \arg\max_k \frac{1}{|\mathcal{L}|} \sum_{l \in \mathcal{L}} \hat{P}_l(\boldsymbol{z}_{i,T}^{(k)}) = k_i^*. \tag{14}$$

Note that the average accuracy across all positions may not be the same as the objectives. Nevertheless, it would be a good proxy for them.

## 5 ITRSA++

As introduced in subsection 2.1, recurrent models, such as AKOrN and HRM, incorporate various techniques that may be redundant with C-voting. To verify if higher performance can be achieved with a simpler and optimized model for C-voting, we introduce ItrSA++, a simple recurrent model to which C-voting can be applied. The design principles of ItrSA++ are as follows: (1) Cross-attention is employed to mix random initialization with input, similar to Perceiver (Jaegle et al., 2021). As in (Geiping et al., 2025), a linear layer was also tested, but experiments showed that cross-attention performed better. (2) Rather than recurrent transformers, we adopt recurrent attention layers to maintain a simple architecture. Experiments showed that periodically inserting SwiGLU (Shazeer, 2020) outperforms, so we adopt it. (3) The normalization method was determined experimentally. As noted in Geiping et al. (2025), normalization has a significant impact on the performance. Appendix D confirms these design choices.

Concretely, ItrSA++ is defined as follows. At first, we embed the input $\boldsymbol{x}_i$ using an embedding layer and apply normalization.

$$\boldsymbol{x}_i^{\text{emb}} = \text{Norm}_{\text{attn}}(\text{Embedding}(\boldsymbol{x}_i)). \tag{15}$$

Hereafter, we use RMSNorm (Zhang & Sennrich, 2019) for the normalization. The initial latent state is sampled from a standard normal distribution. To mix the information of the input and the latent state, we use cross-attention as follows;

$$\tilde{\boldsymbol{z}}_{i,t} = \text{Norm}_{\text{attn}}(\boldsymbol{z}_{i,t}) + \text{CrossAttn}(q = \text{Norm}_{\text{attn}}(\boldsymbol{z}_{i,t}), k = \boldsymbol{x}_i^{\text{emb}}, v = \boldsymbol{x}_i^{\text{emb}}). \tag{16}$$

After mixing the information, we apply the self-attention layer $S \in \mathbb{N}$ times.

$$\bar{z}_{i,t,s+1} = \text{Norm}_{\text{attn}}(\bar{z}_{i,t,s}) + \text{SelfAttn}(\text{Norm}_{\text{attn}}(\bar{z}_{i,t,s})), \quad \bar{z}_{i,t,0} = \tilde{z}_{i,t}, \quad (17)$$

At the end of the iterative self-attention, we apply SwiGLU (Shazeer, 2020) and obtain

$$z_{i,t+1} = \text{Norm}_{\text{mlp}}(\bar{z}_{i,t+1,S}) + \text{SwiGLU}(\text{Norm}_{\text{mlp}}(\bar{z}_{i,t+1,S})). \quad (18)$$

By repeating a block composed of equation 16, equation 17, and equation 18 $T$ times, we obtain the final latent state $z_{i,T}$. The readout module consists of a normalization and a linear layer, defined by

$$\text{logit}_i = W_O \, \text{Norm}_{\text{out}}(z_{i,T}). \quad (19)$$

We use Geometry-Aware Attention Mechanism (Miyato et al., 2024) as the position embedding.

## 6 EXPERIMENTS

### 6.1 BENCHMARKS

Sudoku is a logical puzzle in which the digits 1 to 9 are placed in $9 \times 9$ cells. The cells must be filled with digits to satisfy the following constraints: no digit appears more than once in any row, any column, and any of the nine $3 \times 3$ sub-grids. Several cells are filled with digits in advance as clues, and players fill in the digits referencing these clues. The difficulty of the Sudoku depends on the number of given digits and their placement. We adopt three kinds of datasets: Sudoku, Sudoku-hard, and Sudoku-extreme datasets. Sudoku and Sudoku-hard datasets are identical to those defined in (Miyato et al., 2025). Sudoku dataset (Wang et al., 2019) consists of 10,000 boards with (31–42) given digits. The first 9,000 samples are extracted for training, and the remaining 1,000 samples are extracted for validation. On the other hand, the Sudoku-hard dataset (Palm et al., 2018) contains (17–34) given digits, which is much fewer than the Sudoku dataset. Sudoku-extreme is introduced in (Wang et al., 2025) and is a challenging dataset. It is composed of Sudoku problems from multiple datasets. We use 1,000 samples for training and 1,000 samples for testing and apply the same data augmentation for Sudoku-extreme datasets as in (Wang et al., 2025).

Maze-hard is also a logical puzzle in which the players find the optimal path given $30 \times 30$ cells with passable and impassable cells, a start cell, and a goal cell. We use the same dataset as in (Wang et al., 2025), which contains 1,000 samples in both the training and test datasets.

For the metric of the model performance, we use the board accuracy, which is defined as the proportion of test instances for which the model produces an entirely correct board, i.e., all cells satisfy the given constraints without any errors.

### 6.2 C-VOTING VS. E-VOTING IN AKORN

To compare the performance of C-voting and E-voting, we conduct test-time scaling experiments using AKOrN. For the inference, we use C-voting instead of E-voting and run E-voting on the same pre-trained model. Note that no modification is needed to integrate C-voting into AKOrN.

Figure 2 shows that the board accuracy under C-voting outperforms that of E-voting as the number of random samples increases. For Sudoku-hard test in Figure 2(a), we use the Sudoku dataset for training and the Sudoku-hard dataset for inference during testing. This experimental setting is identical to that in (Miyato et al., 2025). The board accuracy in C-voting with 4096 samples is $94.4 \pm 0.1\%$, which is $4.9\%$ higher than that of E-voting ($89.5 \pm 2.5\%$) reported in (Miyato et al., 2025). We also compare the performance with the Sudoku-extreme dataset in Figure 2(b). This result indicates that C-voting outperforms E-voting and is useful for models even with an explicit energy function. For Maze-hard, AKOrN fails to learn, resulting in a test board accuracy of 0.

### 6.3 ITRSA++ WITH C-VOTING

To show the performance of ItrSA++ with C-voting, we train ItrSA++ for Sudoku, Sudoku-extreme, and Maze-hard datasets and compare the test-time accuracy to HRM and AKOrN.

The results are shown in Figure 3. Even without the voting, ItrSA++ outperforms AKOrN and HRM for all the tasks. Note that for the HRM experimental results for Sudoku-extreme and Maze-hard,

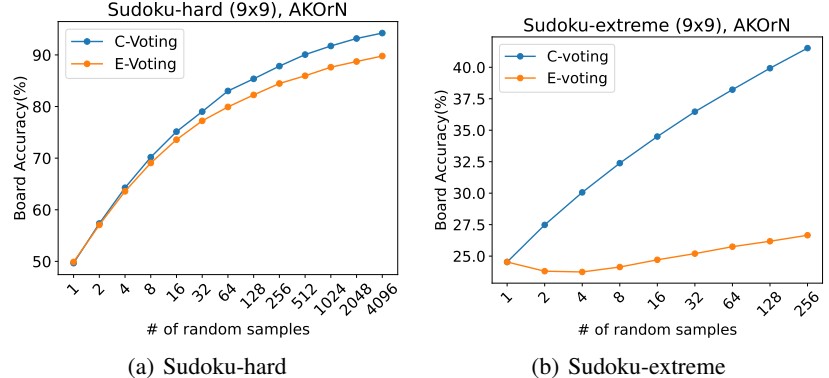

Figure 2: A performance comparison between C-voting and E-voting in AKOrN.

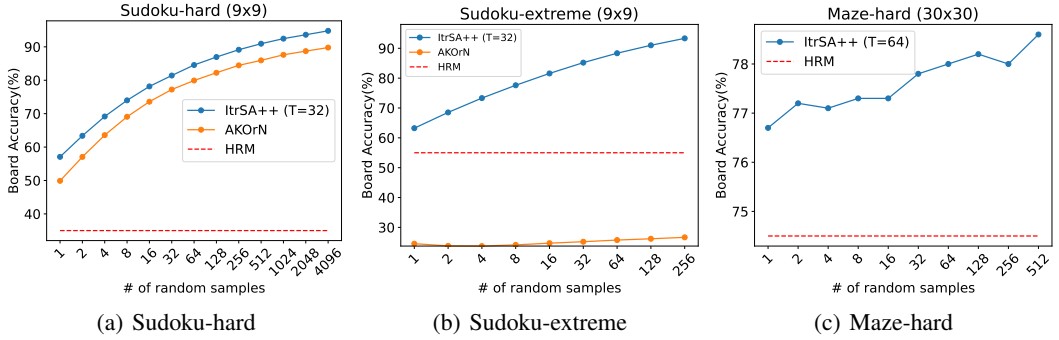

Figure 3: Scaling analysis of ItrSA++ with C-voting.

we utilize those reported in (Wang et al., 2025). On the other hand, for the HRM results on Sudoku-hard, we employ the original HRM code and modify only the dataset. For all of the AKOrN results, we employ the original code (Miyato et al., 2025) and implement datasets for Sudoku-extreme and Maze-hard. As mentioned in subsection 6.2, the test board accuracy of AKOrN for Maze-hard is 0 and not plotted in the figure. ItrSA++ with C-voting scales similarly to AKOrN for the Sudoku tasks when the number of random samples increases, and also exhibits similar scaling for the Sudoku-extreme task. For Maze-hard, the improvement with scaling is weaker compared to Sudoku-hard or Sudoku-extreme, but it nonetheless surpasses HRM and confirms the effectiveness of the voting.

## 6.4   HRM WITH C-VOTING

To demonstrate the general applicability of C-voting, we combine C-voting with HRM and evaluate the scaling ability on the Sudoku-extreme dataset. While HRM initializes the latent state with random variables, it fixes throughout training and inference. To integrate C-voting with HRM, we modify to use a different initial latent state for each input. We also set the exploration rate for ACT to 1, which means that ACT is not actually used. With these modifications, we train HRM and use C-voting at test-time.

Figure 4 shows the results for different numbers of random samples. The light blue solid line represents the board accuracy of HRM with C-voting, and the red dotted line represents that of the original HRM reported in (Wang et al., 2025). Increasing the number of initial random samples demonstrates improved performance. Note in our experiments, we extended HRM to support randomized initial states, while HRM is originally designed to use a fixed initial state across all samples, where introducing randomness can destabilize its updates. Thus, while C-voting shows clear improvements for AKOrN and ItrSA++, its effect on HRM is limited under this modified setting. Nevertheless, even with this mismatch, we observe non-trivial gains when increasing the number of samples.

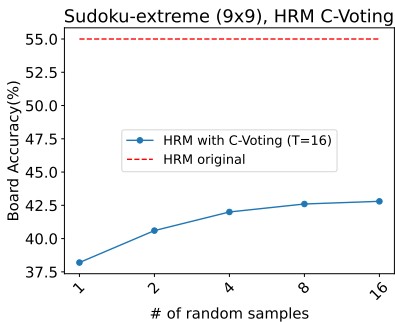

Figure 4: Board accuracy for Sudoku-extreme task of HRM with C-voting.

While the computational complexity of C-voting increases linearly with the number of votes, the computational complexity increase in HRM when increasing the maximum ACT step is expected to be sublinear. Note, however, that our results demonstrate C-voting can be scaled in a manner distinct from scaling in the direction of recurrence and does not focus on computational complexity.

### 6.5 Visualization of confidence

To analyze the reasons why the performance doesn't improve much on the Maze dataset compared to the Sudoku dataset, where it scales nicely, we visualize the confidence in this section.

First, we show the reliability diagram with several temperatures and the relationship between the expected calibration error (ECE) and average accuracy across all positions with 16 different initial states in Figure 5. When varying the temperature, the ranking of the probability for each position remains unchanged, but the average confidence slightly changes, allowing candidate rankings to fluctuate a little. In fact, calibration does change, with ECE reaching a minimum at T=2. However, we also observe that the impact of calibration on average accuracy across all positions is limited. In our setting, C-voting relies on the relative ordering of confidence among sampled candidates, rather than the absolute calibration measured by ECE. Temperature scaling changes the absolute confidence values but preserves the top-1 ordering within each cell, explaining why ECE varies while accuracy remains nearly unchanged.

We show the distributions of confidence for 32 samples with 32 different initial states grouped by whether predictions are correct in Figure 6. For the Sudoku-extreme samples, we observe a broader distribution for incorrect samples than for correct samples. This indicates that when problems are difficult, predictions become unstable due to the randomness of the initialization. On the other hand, the lower histogram, which is from the Maze-hard samples, shows that even for the incorrect predictions, the distribution is tight. This suggests that the model possesses incorrect confidence in the Maze-hard samples, regardless of its initial state.

Figure 7 shows the relationship between the number of iteration steps and confidence for 32 random initial states. In all cases, it can be seen that confidence increases as the number of iteration steps increases. For the Sudoku-extreme dataset, the confidence is low when predictions are incorrect. In contrast, this trend is not observed in Maze-hard, which again indicates that the model has incorrect confidence for incorrectly-predicted samples in the Maze-hard dataset.

## 7 Discussion and Conclusion

We propose C-voting, a new test-time voting strategy applicable even when the explicit energy function is not defined. It starts from multiple initial random latent states in models with recurrent structures and selects the best candidate based on confidence.

It provides test-time scaling other than increasing recursion depth. We show that C-voting can be integrated into AKOrN, and find that it outperforms E-voting in subsection 6.2. Although the reason

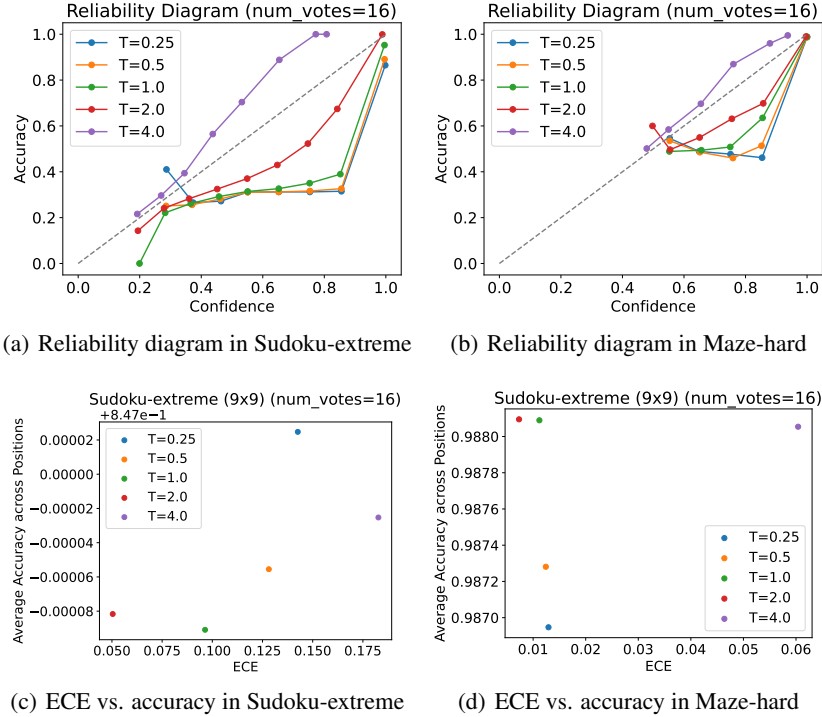

(a) Reliability diagram in Sudoku-extreme     (b) Reliability diagram in Maze-hard

(c) ECE vs. accuracy in Sudoku-extreme     (d) ECE vs. accuracy in Maze-hard

Figure 5: Visualization of calibration of ItrSA++.

for the improvement is not yet clear, at least in the Sudoku task, the confidence may be a more direct proxy for board accuracy than energy.

C-voting achieves state-of-the-art performance by integrating with a new lightweight recurrent model. We introduce ItrSA++ in section 5 as a simple but powerful example of recurrent models for which C-voting can be applied. In subsection 6.3, we demonstrate that ItrSA++ integrated with C-voting outperforms AKOrN in Sudoku tasks, and HRM in Sudoku and Maze-hard tasks.

For models such as AKOrN and ItrSA++, which are trained to operate with random initializations, the sampled candidates evolve into meaningfully different predictions, making C-voting effective. For HRM, however, the original design relies on a fixed initial state shared across all inputs. Introducing randomness at inference time breaks this assumption and can lead to similar or unstable hidden-state evolutions across samples, which fundamentally limits the improvement obtainable from C-voting. Nevertheless, even under this mismatch, we still observe non-trivial gains when the number of samples increases.

As seen in subsection 6.5, performance gains from voting are limited when the model holds incorrect confidence or makes similar predictions regardless of initial random variables. On the other hand, it would also be possible that even if the prediction accuracy of the models cannot be improved, C-voting may still enhance performance by making the models produce more diverse outputs.

Since voting based on confidence corresponds to the maximization of average accuracy across all positions, as shown in equation 14, it would be useful for masked language models (Devlin et al., 2019; Joshi et al., 2020; Li et al., 2022), text infilling (Zhu et al., 2019; Raffel et al., 2020), or image inpainting (Lugmayr et al., 2022; Pathak et al., 2016). Furthermore, using other uncertainty metrics, such as entropy or the sum of log probabilities, may potentially improve performance for specific tasks. We would like to address these as topics for future research.

**Reproducibility Statement** We provide anonymized codes containing training and evaluation scripts for HRM with C-voting, AKOrN with C-/E-voting, and ItrSA++. We specify all hyper-parameters in Tables 1–3, including optimizer settings, gradient clipping, EMA, iteration steps T,

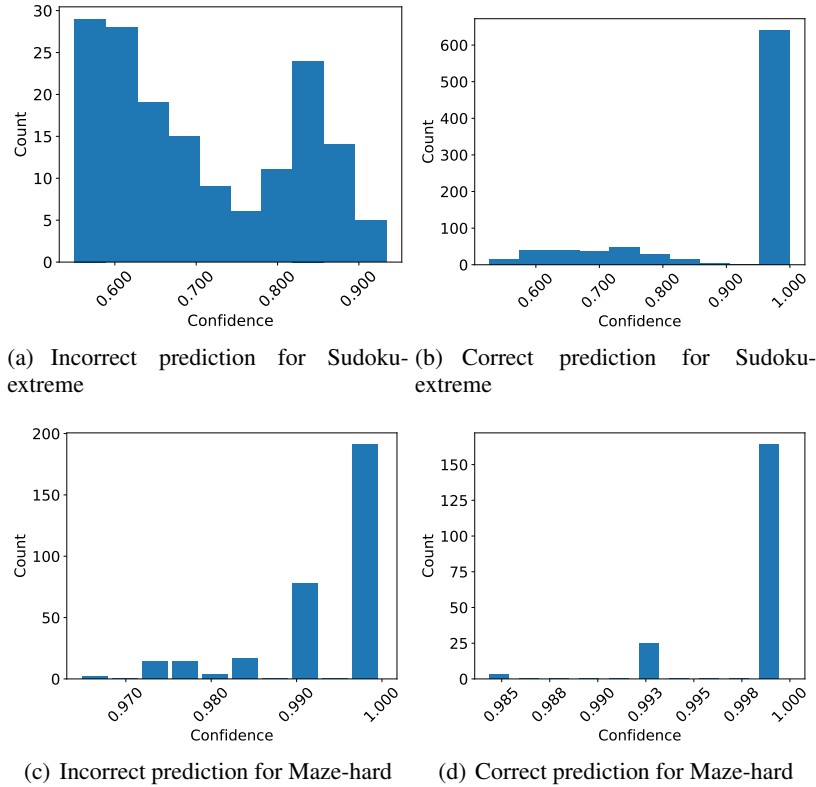

(a) Incorrect prediction for Sudoku-extreme

(b) Correct prediction for Sudoku-extreme

(c) Incorrect prediction for Maze-hard

(d) Correct prediction for Maze-hard

Figure 6: Distributions of confidence of ItrSA++ at $t = 32$.

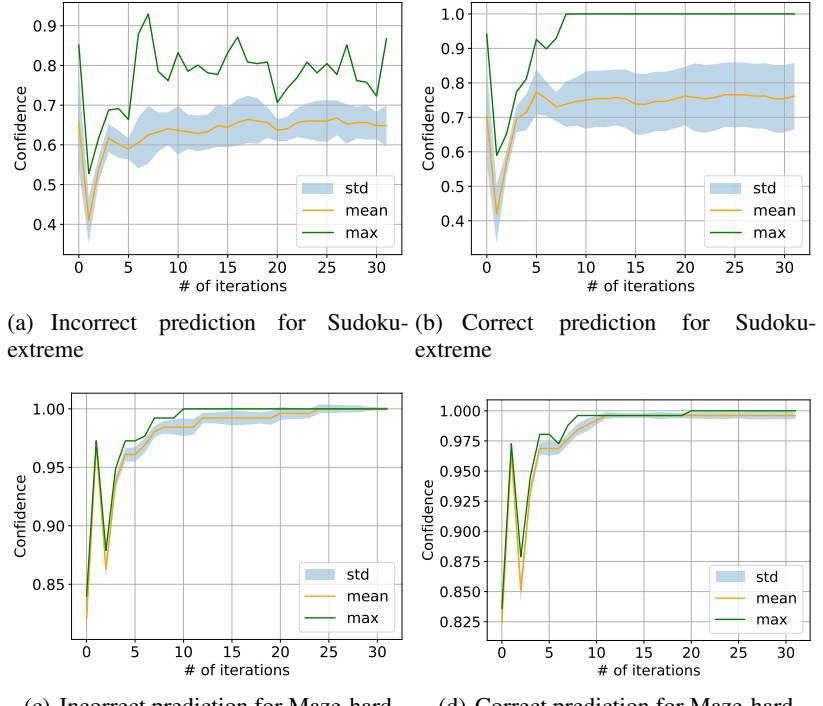

(a) Incorrect prediction for Sudoku-extreme

(b) Correct prediction for Sudoku-extreme

(c) Incorrect prediction for Maze-hard

(d) Correct prediction for Maze-hard

Figure 7: Time step dependency of confidence.

and the number of attention heads. We also provide exact data preprocessing and augmentation pipelines for Sudoku, Sudoku-hard, Sudoku-extreme, and Maze-hard.

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

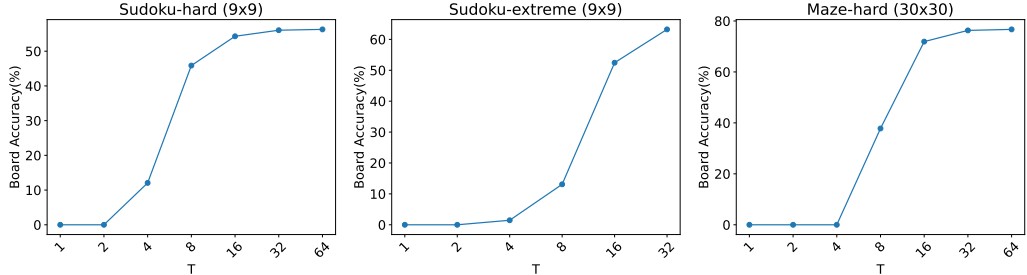

Figure 8: Test-time scaling in ItrSA++

Xuezhi Wang, Jason Wei, Dale Schuurmans, Quoc V Le, Ed H. Chi, Sharan Narang, Aakanksha Chowdhery, and Denny Zhou. Self-consistency improves chain of thought reasoning in language models. In *The Eleventh International Conference on Learning Representations*, 2023. URL https://openreview.net/forum?id=1PL1NIMMrw.

Shunyu Yao, Dian Yu, Jeffrey Zhao, Izhak Shafran, Tom Griffiths, Yuan Cao, and Karthik Narasimhan. Tree of thoughts: Deliberate problem solving with large language models. *Advances in neural information processing systems*, 36:11809–11822, 2023.

Biao Zhang and Rico Sennrich. Root mean square layer normalization. *Advances in Neural Information Processing Systems*, 32, 2019.

Denny Zhou, Nathanael Schärli, Le Hou, Jason Wei, Nathan Scales, Xuezhi Wang, Dale Schuurmans, Claire Cui, Olivier Bousquet, Quoc Le, et al. Least-to-most prompting enables complex reasoning in large language models. *arXiv preprint arXiv:2205.10625*, 2022.

Wanrong Zhu, Zhiting Hu, and Eric Xing. Text infilling. *arXiv preprint arXiv:1901.00158*, 2019.

## A   TEST-TIME SCALING IN ITRSA++

ItrSA++ has recursive structures similar to models such as the recurrent transformer (Geiping et al., 2025; Jaegle et al., 2021), and test-time scaling can be observed. Figure 8 demonstrates that for Sudoku-hard, Sudoku-extreme, and Maze-hard tasks, board accuracy increases as the number of iterative steps grows in ItrSA++.

## B   TRANSFORMER WITH C-VOTING

To validate the performance improvement by C-voting for other than recurrent models, we also integrate it into a transformer and conduct experiments. Training is performed using the Sudoku dataset, and testing is performed using the Sudoku-hard dataset. This is the same problem setting as subsection 6.2. Unlike a standard transformer, it requires a random initial latent state to use C-voting. Therefore, we mix the input $x$ and the initial latent state $z$ using a linear layer before feeding them into the transformer.

Figure 9 shows the dependence of board accuracy on the number of random samples for initial states. Since the transformer is not performing well on the Sudoku-hard task, performance improvements through C-voting are also limited.

## C   EXPERIMENTAL DETAILS

In this section, we provide an overview of the experimental details. We apply exponential moving average (Karras et al., 2023; Lee et al., 2024; Li et al., 2024) for the model parameters of ItrSA++ and AKOrN. We adopt gradient truncation (Geiping et al., 2025) for ItrSA++. More precisely, during the training, we detach the gradient of the latent state $z_t$ at $t = 2$ for Sudoku, at $t = 14$ for

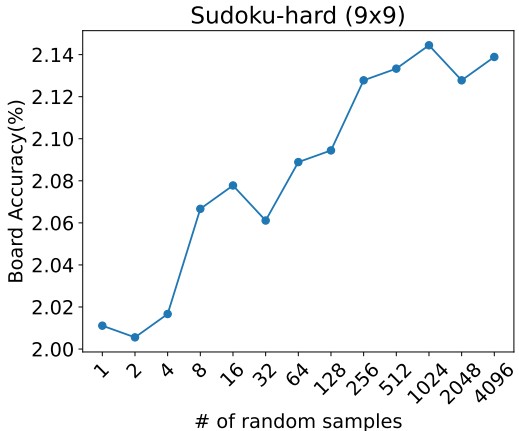

Figure 9: Board accuracy of a transformer with C-voting.

| Parameter | Value |
|---|---|
| Optimizer | AdamW |
| $\beta$ for Adam | $(0.9, 0.95)$ |
| Weight decay | 0.01 |
| Gradient clipping threshold | 1.0 |
| Learning rate | $5 \times 10^{-4}$ |
| Batch size | 64 |
| EMA decay rate | 0.995 |
| # of heads | 12 |
| Embedding dims. | 384 |
| # of repetitions of Self-Attn. | 4 |

Table 1: Hyperparameters for ItrSA++.

Sudoku-extreme, and at $t = 28$ for Maze. We show the hyperparameters for ItrSA++ in Table 1, for HRM in Table 2, and for AKOrN in Table 3. The number of parameters is $\approx$3 million for AKOrN and ItrSA++, and $\approx$27 million for HRM with these settings.

| Parameter | Value |
|---|---|
| Optimizer | Adam-atan2 (Everett et al., 2024) |
| $\beta$ for Adam | $(0.9, 0.95)$ |
| Weight decay | 1.0 |
| Gradient clipping threshold | 1.0 |
| Learning rate | $1 \times 10^{-4}$ |
| Warm-up steps | 2000 |
| Batch size | 768 |
| # of heads | 8 |
| Embedding dimension | 512 |
| Epochs | 20000 |
| # of H layers | 4 |
| # of L layers | 4 |
| Halt exploration prob. | 1.0 |

Table 2: Hyperparameters for modified HRM.

| Parameter | Value |
|---|---|
| Optimizer | Adam |
| $\beta$ for Adam | $(0.9, 0.999)$ |
| Gradient clipping threshold | 1.0 |
| Learning rate | $5 \times 10^{-4}$ |
| Batch size | 100 |
| EMA decay rate | 0.995 |
| # of heads | 8 |
| Embedding dimension | 512 |
| Epochs | 100 |
| # of iteration steps | 16 |
| # of Kuramoto layers | 1 |
| Oscillator dims. | 4 |
| Step size | 1.0 |
| Connectivity | Attention |

Table 3: Hyperparameters for AKOrN.

| Variant | Board acc. in test |
|---|---|
| w/o GT | 0.0% |
| w/o EMA | 0.0% |
| Cross attn. to linear | 59.2% |
| SWiGLU to FFN | 62.0% |
| ItrSA++(Proposed) | **63.2%** |

Table 4: Ablation study for ItrSA++

# D    ABLATION STUDY OF ITRSA++

To clarify the contribution of each architectural and inference component of ItrSA++, we conduct ablation experiments on Sudoku-extreme. Table 4 summarizes the results.

Overall, the ablations demonstrate that (1) ItrSA++ requires EMA and GT for stable training, (2) the cross-attention block provides meaningful improvements in reasoning ability.

We further examine how the effectiveness of voting changes when using metrics other than top-1 probability $\hat{P}_l(z_{i,T}^{(k)})$ in Equation 9. Specifically, we use negative entropy (NE) and the sum of log probability (LP) defined below.

$$\hat{P}_l^{(\text{NE})}(z_{i,T}^{(k)}) = \sum_j P_{j,l}(z_{i,T}^{(k)}) \log P_{j,l}(z_{i,T}^{(k)}) \tag{20}$$

$$\hat{P}_l^{(\text{LP})}(z_{i,T}^{(k)}) = \max_j \log P_{j,l}(z_{i,T}^{(k)}) \tag{21}$$

The results with ItrSA++ are shown in Figure 10. For Sudoku-extreme, almost no difference is observed, and even in Maze-hard, the difference is only about 0.1%. This is thought to be because when the top-1 probability is dominant, there is little difference in ranking across metrics. On the other hand, since it facilitates analyses such as Equation 14, we adopt the top-1 probability.

# E    PERFORMANCE DEPENDENCY ON RANDOM SEEDS

Our method uses random variables, so performance may vary when different random seeds are used. To verify this, we perform inference on ItrSA++ with C-voting using multiple random seeds. Figure 11 shows the box plots of the results obtained using five different seeds. Compared with Sudoku-extreme, Maze-hard exhibits greater seed dependence, though it remains at around 1%. However, looking at the median, board accuracy increases almost monotonically, indicating that the overall results are not significantly affected.

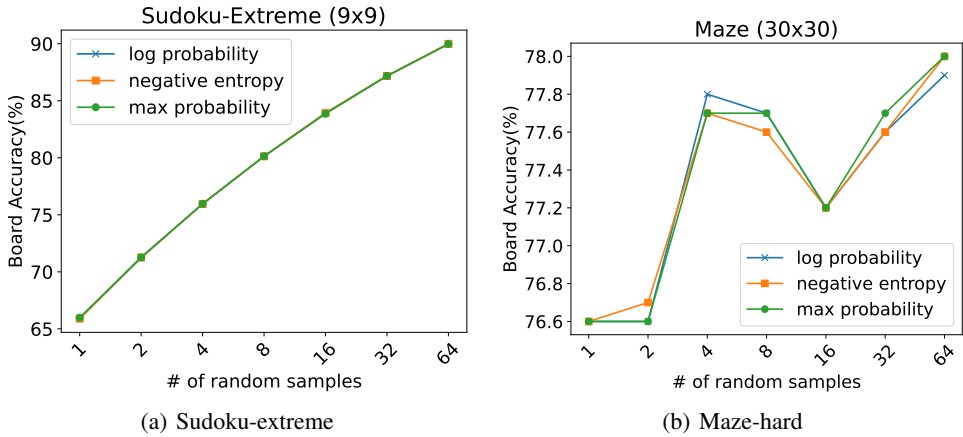

Figure 10: Comparison of voting effects across different metrics.

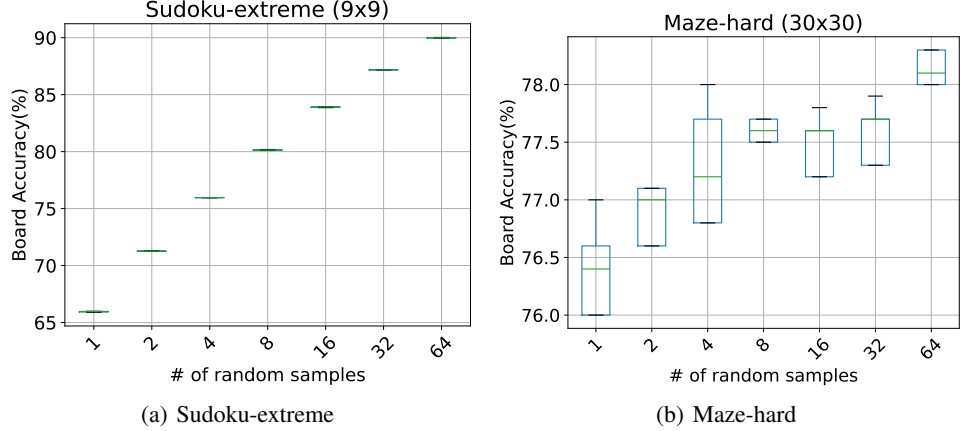

Figure 11: Random seed dependency of board accuracy.

## F USE OF LARGE LANGUAGE MODELS

We use large language models (LLMs) to refine writing for this paper and partially use them for code generation of experiments. We also use LLMs to help explore related works.

