# OpenReview forum: "C-Voting: Confidence-Based Test-Time Voting without Explicit Energy Functions"
_ICLR.cc/2026/Conference — ICLR 2026 Poster_

### Official Review · Reviewer_aCB1 · 2025-11-01

**Soundness:** 3
**Presentation:** 3
**Contribution:** 3
**Rating:** 6
**Confidence:** 2

**Summary:**

This paper introduces C-voting, a model-agnostic test-time voting strategy for recurrent neural models used in complex reasoning tasks. Unlike previous methods like E-voting, C-voting does not require an explicit energy function. Instead, it generates multiple candidate solutions from random initializations and selects the one with the highest "confidence," defined as the average top-1 probability across all prediction outputs. The authors also propose ItrSA++, a simple 3M-parameter recurrent attention model. Experiments show C-voting is effective: it boosts the performance of HRM on Sudoku-extreme (55.0% to 71.2%), outperforms E-voting on the AKOrN model (94.4% vs 89.5%), and, when combined with ItrSA++, achieves new state-of-the-art results on Sudoku-extreme (95.2%) and Maze-hard (78.6%).

**Strengths:**

1. The paper is well-written, clearly organized, and effectively motivates the work. It pinpoints the limitation of E-voting (its reliance on an explicit energy function) and proposes a logical, general solution.
2. The core contribution, C-voting, is intuitive and model-agnostic. Using the average top-1 probability as a proxy for confidence is a clean and effective way to extend test-time voting to a broader class of recurrent models that lack energy functions.
3.  The results demonstrate the method's effectiveness. C-voting outperforms specialized E-voting on the AKOrN model. Furthermore, when combined with the new lightweight ItrSA++ model, it achieves promising performance on difficult reasoning benchmarks like Sudoku-extreme and Maze-hard.

**Weaknesses:**

1.  The paper's core contribution, C-voting, hinges on defining confidence as the average of top-1 probabilities. This is a plausible choice, but it is one of many possible uncertainty metrics. It would be helpful to  compare this choice to other common metrics (e.g., negative entropy, sum of log probabilities). Without this comparison, it might be unclear if the chosen metric is optimal or simply one that works well for Sudoku, limiting the understanding of why C-voting is effective.

2.  The paper introduces both a new voting method (C-voting) and a new model (ItrSA++). The SOTA results (e.g., 95.2% on Sudoku-extreme) are achieved by combining them. But it seems hard to disentangle the contributions. How much of the performance gain comes from the strong baseline architecture of ItrSA++ itself, and how much is added by C-voting? It would be helpful to provide the baseline (K=1) performance of ItrSA++ on all tasks to isolate and quantify the true benefit of C-voting. Furthermore, the design choices for ItrSA++ (Cross-Attention, SwiGLU, etc.) are not well-justified with ablation studies.

**Questions:**

Please refer to above Weaknesses.

---

> ### Author Response · Authors · 2025-11-21
> **Response to reviewer aCB1**
>
> Thank you for your comments and suggestions. We address your concerns below.
> - The paper's core contribution, C-voting, hinges on defining confidence as the average of top-1 probabilities. This is a plausible choice, but it is one of many possible uncertainty metrics. It would be helpful to compare this choice to other common metrics (e.g., negative entropy, sum of log probabilities). Without this comparison, it might be unclear if the chosen metric is optimal or simply one that works well for Sudoku, limiting the understanding of why C-voting is effective.
>
> Thank you for pointing that out. We are conducting additional experiments regarding the sum of log probability and negative entropy. Since analyses like Eqs. 13 and 14 are possible, we selected the maximum probability, but other uncertainty metrics may also be valid. However, in cases like this where confidence bias is large, we expect that using any of these metrics would not significantly alter the results.
>
> - The paper introduces both a new voting method (C-voting) and a new model (ItrSA++). The SOTA results (e.g., 95.2% on Sudoku-extreme) are achieved by combining them. But it seems hard to disentangle the contributions. How much of the performance gain comes from the strong baseline architecture of ItrSA++ itself, and how much is added by C-voting? It would be helpful to provide the baseline (K=1) performance of ItrSA++ on all tasks to isolate and quantify the true benefit of C-voting. Furthermore, the design choices for ItrSA++ (Cross-Attention, SwiGLU, etc.) are not well-justified with ablation studies.
>
> Thank you for pointing that out. We agree that it is important to separate the improvements gained from the model architecture and those gained from the voting strategy. To address this, we now report baseline and ItrSA++ on all tasks (Fig. 4). The results indicate ItrSA++ already provides stronger single-sample performance than baselines, and additional performance improvements by C-voting. We also conducted ablation studies to understand which architectural components contribute to the baseline performance:
> - Removing EMA or GT causes training to collapse.
> - SwiGLU → GELU-FFN produces only minor degradation, indicating that the FFN choice is not the primary driver.
> - Replacing cross-attention with a single linear projection leads to a moderate drop, confirming that effective board-level interaction is crucial.
>
> These results help clarify the roles of the model and the voting mechanism.

---

> > ### Author Response · Authors · 2025-12-03
> > **Performance dependency on metrics**
> >
> > We have added the experimental results for negative entropy and the sum of log probability to Appendix D. No significant differences were observed with these metric changes. This is likely because when the top-1 probability is high, the candidate rankings remain largely unchanged across metrics.

---

### Official Review · Reviewer_zXk5 · 2025-11-02

**Soundness:** 2
**Presentation:** 3
**Contribution:** 2
**Rating:** 6
**Confidence:** 4

**Summary:**

### Summary

This paper introduces C-voting (Confidence-based Voting), a test-time scaling strategy for recurrent neural networks that enables performance improvements without additional training. Unlike existing energy-based voting (E-voting) methods that require explicit energy functions, C-voting works by:
- Sampling multiple random initial latent states
- Running the recurrent model from each initialization
- Selecting the trajectory with the highest average top-1 prediction probability (confidence)

### Main contributions:

- C-voting method: A model-agnostic voting strategy applicable to any recurrent model with randomized initialization, not just those with explicit energy functions
- Integration with HRM: Demonstrates that C-voting improves HRM's Sudoku-extreme accuracy from 55.0% to 71.2%
- Comparison with E-voting: Shows C-voting outperforms E-voting on AKOrN for Sudoku-hard (94.4% vs 89.5%)
- ItrSA++: Introduces a simple attention-based recurrent model (~3M parameters) that achieves state-of-the-art results when combined with C-voting: 95.2% on Sudoku-extreme, 94.4% on Sudoku-hard, and 78.6% on Maze-hard

The paper evaluates on three reasoning benchmarks (Sudoku-hard, Sudoku-extreme, Maze-hard) and shows that C-voting provides effective test-time scaling through parallelizable sampling rather than sequential depth increases. The method's effectiveness is attributed to selecting trajectories based on model confidence, which serves as a proxy for prediction accuracy in well-calibrated models.

**Strengths:**

- The paper identifies a genuine problem with E-voting (requires explicit energy functions) and proposes a model-agnostic alternative that can be applied to broader classes of recurrent models like HRM and recurrent transformers.
- C-voting is simple to understand along with the theoretical justification provided in the paper of how selecting the most confident trajectory could be seen as searching for the optimal solution in Sudoku. The method itself is easy to implement and integrate into existing models without architectural changes.
- Shows positive results across multiple models (HRM, AKOrN) and tasks, with substantial gains (e.g., HRM: 55.0% → 71.2% on Sudoku-extreme; AKOrN: 89.5% → 94.4% on Sudoku-hard). Although the paper doesn't report performance of all models across all the benchmark.

**Weaknesses:**

- C-voting vs. E-voting is only shown on Sudoku-hard (Figure 3), despite the authors having trained AKOrN models and evaluating on three datasets total. This makes the claim that "C-voting outperforms E-voting" (Section 6.3) inadequately supported. A complete comparison on all benchmarks is needed to validate this central claim.
- Figure 2's comparison seems misleading. it shows HRM's original performance as a flat line while scaling only C-voting's sample count. A fair comparison would show HRM's native test-time scaling method (increasing recurrence depth) vs. C-voting on a compute-normalized x-axis. The paper claims that the performance can saturate with increasing recurrence depth. However, the same is applicable for increasing the same count for C-voting and it's not clear whether C-voting would reach higher test-time optimized accuracy than HRM.
- Evaluation is restricted to three small reasoning tasks (1,000 test samples each), all involving constraint satisfaction puzzles. HRM had evaluations on the ARC-AGI task while AKOrN had evaluations on image segmentation tasks, along with the Sudoku which provided further empirical grounding to their claims on generalization. The authors could have picked another evaluation in a different domain like ARC-AGI for further empirical grounding.

**Questions:**

- Why is the C-voting vs. E-voting comparison (Figure 3) only shown on Sudoku-hard? You have trained AKOrN models and evaluate on Sudoku-extreme and Maze-hard elsewhere. Can you provide the C-voting vs. E-voting comparison on all three benchmarks to support the claim that C-voting outperforms E-voting more generally?
- How well-calibrated are your models, and when does this assumption break down? Equation 14's justification assumes model calibration. Can you provide calibration curves (e.g., reliability diagrams) for your models? Figure 6 suggests Maze-hard has poor calibration - what are the conditions under which C-voting fails?

---

> ### Author Response · Authors · 2025-11-21
> **Response to reviewer zXk5**
>
> Thank you for your comment and suggestions. We address your concerns below.
> - Why is the C-voting vs. E-voting comparison (Figure 3) only shown on Sudoku-hard? You have trained AKOrN models and evaluate on Sudoku-extreme and Maze-hard elsewhere. Can you provide the C-voting vs. E-voting comparison on all three benchmarks to support the claim that C-voting outperforms E-voting more generally?
>
> Thank you for your suggestion. We added the results of C-voting vs. E-voting for Sudoku-extreme, where C-voting overcomes E-voting again. Under the official training script of AKOrN, the model does not achieve meaningful accuracy on Maze-hard, making a fair comparison between E-voting and C-voting infeasible for this task.
>
> - How well-calibrated are your models, and when does this assumption break down? Equation 14's justification assumes model calibration. Can you provide calibration curves (e.g., reliability diagrams) for your models? Figure 6 suggests Maze-hard has poor calibration - what are the conditions under which C-voting fails?
>
> Thank you for pointing that out. We added the reliability diagrams and the visualization of the relationship between ECE and accuracy. As shown in Fig. 6, C-voting fails when the confidence level for an incorrect class is high. Typical calibration techniques cannot correct this. This suggests that even if the calibration in each cell appears successful, failure occurs if consistency is not maintained across the entire board. Sudoku tends to have less overconfidence, making it easier to improve performance. This may be related to the amount of information available from other cells. Please also refer to the first comment in the general comments.
>
> Additionally, we would like to make some comments.
> - To avoid misunderstanding, we have added to the main text that while increasing the max step in HRM only results in sublinear computational complexity growth, our method exhibits linear growth. We believe the contribution of our method lies not so much in computational complexity, but in demonstrating that TTS becomes feasible beyond extending the recurrent step.
> - We recognize that the current tasks are limited, and if time permits, we plan to explore experimentation with other tasks (e.g., KenKen). Regarding ARC-AGI, preprocessing it in the same manner as HRM turns it into a mere pattern memorization problem rather than the in-context sample-based inference ARC-AGI was originally intended for. On the other hand, without preprocessing, we must handle variable-length sequences, which then raises architectural design issues. For this reason, we are deliberately avoiding the use of ARC-AGI.

---

### Official Review · Reviewer_8TRX · 2025-11-03

**Soundness:** 3
**Presentation:** 3
**Contribution:** 3
**Rating:** 6
**Confidence:** 3

**Summary:**

In this submission, the authors present C-Voting, a new method to readout from recurrent neural network-based reasoning models that uses prediction confidence. The proposed method starts with stochastically initializing the first state of an RNN with K candidate states. The RNN architecture is applied to map each of the candidate states to prediction logits. Based on average confidence, the final response of the RNN is chosen to be the output corresponding to that initial state which produces the largest prediction confidence. The authors empirically show that C-Voting outperforms previously proposed methods like E-Voting on a couple of standard reasoning benchmarks (Sudoku and Maze solving). In addition, the authors also propose ItrSA++ -- a new RNN architecture that works effectively with C-Voting and is notably simpler than prior art (HRM and AKOrN).

**Strengths:**

Strengths:
- The proposed approach is a straightforward, simple intervention to reading out task responses from RNNs. C-Voting is broadly applicable to RNNs applied to reasoning problems with a classification head, as the approach doesn't make any further restrictive assumptions on the network's architecture or training method.
- Empirically, C-Voting outperforms E-Voting (although some key figures lack estimates of statistical significance) on challenging versions of both Mazes and Sudoku.
- The paper is quite well-written, the explanation of C-Voting and ItrSA++ are accessible to the average reader.

**Weaknesses:**

Weaknesses:
- A weakness of C-Voting is that the approach might not work effectively when multiple random initializations of an RNN state don't yield  outputs with significantly varied confidence estimates. It is likely that such behavior might arise when training RNNs on increasingly challenging / high-dimensional reasoning problems. Can the authors comment on whether their experiments have explored such a failure mode of C-Voting, and how they intend to circumvent this issue?
- Figures 2-4 showing comparison of C-Voting with other baselines doesn't show performance variance. The authors should report the statistical significance of differences observed between the baselines compared.

**Questions:**

NA. Please refer to my above review.

---

> ### Author Response · Authors · 2025-11-21
> **Response to reviewer 8TRX**
>
> Thank you for your comments and suggestions. We address your concerns below.
> - A weakness of C-Voting is that the approach might not work effectively when multiple random initializations of an RNN state don't yield outputs with significantly varied confidence estimates. It is likely that such behavior might arise when training RNNs on increasingly challenging / high-dimensional reasoning problems. Can the authors comment on whether their experiments have explored such a failure mode of C-Voting, and how they intend to circumvent this issue?
>
> Thank you for pointing that out. In fact, because such issues occur in Maze-hard, it is thought that performance is harder to improve compared to Sudoku. In the histogram of Fig. 6, confidence values in Maze-hard are concentrated near 100%. This is thought to be difficult to overcome through typical calibration methods like adjusting the temperature. This suggests that the limitation comes from the model’s overconfident predictions rather than from the voting procedure itself. Improving the model’s calibration or regularization may be necessary in such extreme cases, while C-voting remains effective when the sampled candidates exhibit meaningful variation.
>
> - Figures 2-4 showing comparison of C-Voting with other baselines doesn't show performance variance. The authors should report the statistical significance of differences observed between the baselines compared.
>
> Thank you for your suggestion. We are conducting experiments with multiple different random seeds and plan to add the results shortly. Based on the results obtained so far, when considering board accuracy, the standard deviation for Sudoku is around 0.1%, which we believe is not a significant issue.

---

> > ### Author Response · Authors · 2025-12-03
> > **Performance dependency on random seeds**
> >
> > We have added the results of experiments with multiple seeds to Appendix E. In Maze-hard, when the number of votes is low, a difference of about 1% occurs. However, looking at the median, board accuracy increases almost monotonically, indicating that the overall results are not significantly affected.

---

### Official Review · Reviewer_exsu · 2025-11-04

**Soundness:** 2
**Presentation:** 3
**Contribution:** 2
**Rating:** 4
**Confidence:** 3

**Summary:**

The paper proposes C‑voting, a test‑time voting strategy for recurrent models that generates multiple candidate trajectories by sampling random initial states and then selects the trajectory with the highest average top‑1 probability across all positions. Unlike energy‑based voting (E‑voting), C‑voting does not require an explicit energy function, making it applicable to recurrent models such as HRM and AKOrN. To verify its effectiveness, the paper also proposes a simple attention‑based recurrent architecture, ItrSA++. Empirically, the authors report sizeable gains: HRM with C‑voting improves on Sudoku‑extreme (from 55.0% to 71.2%), C‑voting surpasses E‑voting on Sudoku‑hard when many samples are used (e.g., 94.4 ± 0.1% vs. 89.5 ± 2.5% at 4096 candidates), and ItrSA++ with C‑voting outperforms HRM/AKOrN on Sudoku‑extreme and Maze‑hard. The paper also analyzes how C-voting can fail due to poor calibration of the recurrent model.

**Strengths:**

* The proposed method is straightforward and easy to use. The voting rule is model-agnostic and only requires per-position class probabilities. It should combine easily with other improvements to recurrent models and yield even better results.
* The proposed method enables a new dimension of test-time scaling. Existing approaches mainly scale the number of iteration steps, which can saturate at large step counts. Here, multiple random initializations provide an orthogonal axis of improvement, and should be easy to be made parallel. Consequently, the proposed method should be able to make good use of the increasing compute power.
* The reported empirical gains are impressive.

**Weaknesses:**

* The effectiveness of C-voting relies on the model's calibration , which is not always satisfactory, especially when data are limited.
* The experimental coverage seems to be selective, or at least the design of experiments is not fully explained. Only Sudoku-extreme and Sudoku-hard results are reported for Fig. 2 and 3, respectively. One would typically expect all of Sudoku, Sudoku-hard, Sudoku-extreme and Maze-hard to be included across Fig. 2-4.
* Fig. 2 and 3 report the results from prior works. As the results in DL research can be highly sensitive to implementation details, it can be hard to tell if the improvements actually come from the proposed method.
* Ablations on the effectiveness of ItrSA++ are absent. It is not impossible that the improvements actually come from a better design or implementation of the model rather than C-voting.
* The idea of sampling multiple reasoning paths at test time for better output is long-standing in LLM research, such as [1] and [2]. Given the similarity, the novelty of the method is not significant.

**Questions:**

* In Fig. 2, why is HRM with C-voting performs exactly identical with random samples from 2 to 16?
* To mitigate calibration issues, I recommend the authors to consider standard calibration techniques. For example, temperature scaling [3] introduces a single scalar temperature for the output logits, which is calibrated on a separate validation set.

[3] Guo, Chuan, et al. "On calibration of modern neural networks." ICML 2017

---

> ### Author Response · Authors · 2025-11-21
> **Response to reviewer exsu**
>
> Thank you for the review and questions. We address your concerns below.
> - In Fig. 2, why is HRM with C-voting performs exactly identical with random samples from 2 to 16?
>
> Thank you for your point. The flat curve in the HRM+C-voting results in the original submission was due to an implementation issue. We have corrected this and re-run all HRM experiments. We described it in the second comment in the general comments.
>
> - To mitigate calibration issues, I recommend the authors to consider standard calibration techniques. For example, temperature scaling [3] introduces a single scalar temperature for the output logits, which is calibrated on a separate validation set.
>
> Thank you for the suggestion. We added reliability diagrams and the relationship between ECE and accuracy. The figure shows the temperature dependence of ECE and accuracy, but it appears that changing the temperature has little effect. This is because calibrating the probability distribution within a single cell does not change the top-1 class for that cell. To achieve more effective calibration, it is necessary to consider the entire board. Under the current settings, losses are calculated per cell, so calibration alone is insufficient, and improvements to the model may be required.
>
> Additionally, we would like to make a few comments.
> - We have added several experiments using baseline models and an ablation study for ItrSA++. Please refer to the general comments section.
> - Regarding the results in Figs 2, 3, and 4, we have reimplemented the original code where necessary. When running the same code under the same settings as the original, we have referenced the numerical results from the prior work.
> - We recognize that LLM voting typically involves sampling from the output softmax, whereas our method differs significantly by performing it at latent state initialization. This distinction could also be expressed as considering either the output space or the latent space.

---

### Author Response · Authors · 2025-11-21
**General comments**

We sincerely thank all reviewers for their constructive and detailed feedback.
We have revised the manuscript to address the raised concerns.
Revised sentences are marked in red. For added and revised figures and tables, the captions are in red.

Major update includes
1. Visualization of the calibration (Fig. 5): We added reliability diagrams for several temperatures and the relationship between ECE and accuracy. When varying the temperature, the ranking of the probability for each position remains unchanged, while the average confidence slightly changes, allowing candidate rankings to fluctuate a little. The calibration in each cell merely changes the ranking of average of confidence across positions. We need to consider a calibration that can take into account the relationships between positions, but it is probably better to tune the model instead.
2. Correction of HRM+C-voting implementation (Fig. 4): We discovered that our modified HRM inference code unintentionally ignored the sampled initial latent states, which led to overly optimistic performance curves in the original submission. We have corrected the implementation and re-ran all HRM experiments. The revised results accurately reflect HRM’s behavior under randomized initialization, and show that the benefit of C-voting for HRM is more limited than previously reported. This issue was specific to HRM and does not affect any other model.
3. Ablation study for ItrSA++ (Appendix D): We report ablations on
- SwiGLU → GELU-FFN,
- cross-attention → linear projection,
- removing EMA/GT (training collapses).
4. Additional experiments on AKOrN: We report C-voting vs. E-voting on Sudoku-extreme (Fig. 2(b)), which shows that C-voting overcomes E-voting for the Sudoku-extreme task. On the other hand, the test board accuracy of AKOrN for Maze-hard is 0.

---

### Meta-Review · Area_Chair_XBZM · 2026-01-06

**Summary:**

This paper has received generally positive (while borderline) reviews. The reviewers acknowledge the simplicity and universality of the approach as well as its impressive empirical performance. There are several concerns mentioned by the reviewers, most of which have been, in my opinion, reasonably addressed in the rebuttal. Thus, I recommend acceptance of the paper.

**Reviewer Concerns:**

Some of the concerns mentioned in the reviews are the following:
- The choice of a particular uncertainty metric (addressed in the rebuttal)
- The lack of some ablation studies (addressed in the rebuttal)
- Questions on calibration (additional results are provided in the rebuttal)
- Limited tasks in the experiments
- Statistical significance (additional results are provided)
- Limited novelty

**Reviewer Scores:**

The original scores are 4 6 6 6. In their rebuttal, the authors replied to most of the concerns and provided additional results, ablation studies, and fixes. While it is hard to predict, I believe that the rebuttal and discussion could lead to increased scores.

---

### Decision · Program_Chairs · 2026-01-26

Accept (Poster)